# Effectiveness of a sepsis programme in a resource-limited setting: a retrospective analysis of data of a prospective observational study (Ubon-sepsis)

Suchart Booraphun,[1] Viriya Hantrakun ,[2] Suwatthiya Siriboon,[1] Chaiyaporn Boonsri,[3] Pulyamon Poomthong,[1] Bung-Orn Singkaew,[1] Oratai Wasombat,[1] Parinya Chamnan,[4] Ratapum Champunot,[5] Kristina Rudd,[6,7] Nicholas P J Day,[2,8] Arjen M Dondorp,[2,8] Prapit Teparrukkul,[1] Timothy Eoin West,[7,9] Direk Limmathurotsakul [2,8,10]

**Correspondence to**
Dr Direk Limmathurotsakul;
direk@tropmedres.ac

## ABSTRACT

**Objective** To evaluate the effectiveness of a Sepsis Fast Track (SFT) programme initiated at a regional referral hospital in Thailand in January 2015.

**Design** A retrospective analysis using the data of a prospective observational study (Ubon-sepsis) from March 2013 to January 2017.

**Setting** General medical wards and medical intensive care units (ICUs) of a study hospital.

**Participants** Patients with community-acquired sepsis observed under the Ubon-sepsis cohort. Sepsis was defined as modified Sequential Organ Failure Assessment (SOFA) Score ≥2.

**Main exposure** The SFT programme was a protocol to identify and initiate sepsis care on hospital admission, implemented at the study hospital in 2015. Patients in the SFT programme were admitted directly to the ICUs when available. The non-exposed group comprised of patients who received standard of care.

**Main outcome** The primary outcome was 28-day mortality. The secondary outcomes were measured sepsis management interventions.

**Results** Of 3806 sepsis patients, 903 (24%) were detected and enrolled in the SFT programme of the study hospital (SFT group) and 2903 received standard of care (non-exposed group). Patients in the SFT group had more organ dysfunction, were more likely to receive measured sepsis management and to be admitted directly to the ICU (19% vs 4%). Patients in the SFT group were more likely to survive (adjusted HR 0.72, 95% CI 0.58 to 0.88, p=0.001) adjusted for admission year, gender, age, comorbidities, modified SOFA Score and direct admission to the ICUs.

**Conclusions** The SFT programme is associated with improved sepsis care and lower risk of death in sepsis patients in rural Thailand, where some critical care resources are limited. The survival benefit is observed even when all patients enrolled in the programme could not be admitted directly into the ICUs.

**Trial registration number** NCT02217592.

## Strengths and limitations of this study

► The study hospital used the published framework, SCAN-TEACH-TREAT programme to develop a context-specific quality of care improvement for sepsis in a tropical resource-limited setting.

► The study took advantage of a robust prospective observational study design that strengthened causal inference by providing pre-intervention information, having an appropriate control group from both pre-intervention and post-intervention periods and controlling important confounding factors (ie, the modified Sequential Organ Failure Assessment (SOFA) Score).

► We found that most measured sepsis interventions increased.

► The study did not record dosage of dobutamine, dopamine, epinephrine and norepinephrine, arterial blood gases were rarely performed and the modified SOFA Score (maximum 23) may be lower than the SOFA Score (maximum 24).

► The observational study may have residual confounding factors such as improvement of care and profile of organ failure recognition overtimes.

## INTRODUCTION

Sepsis is defined as life-threatening organ dysfunction caused by a dysregulated host response to infection,[1] and is the primary cause of death from infection, especially if not recognised and treated promptly.[2–4] Sepsis is a major cause of health loss worldwide and is associated with approximately 11 million deaths each year, most of which occur in low/middle-income countries (LMICs).[5] The United Nations World Health Assembly has recognised sepsis as a global health priority and adopted a resolution on improving its worldwide prevention, diagnosis and management.[6] Comprehensive guidelines such as

those developed by the Surviving Sepsis Campaign (SSC) have been associated with reduced mortality in high-income countries,[2–4] but effectiveness of these guidelines in LMICs needs more evaluation.[7–10]

Following the SSC 2012,[11] the Ministry of Public Health Thailand and the Thai Society of Critical Care Medicine developed local recommendations on sepsis based on resource availability and local context.[12] The recommendations suggest that secondary-care and tertiary-care hospitals in the country should develop a Sepsis Fast Track (SFT) so that, on presentation, sepsis patients can be identified, treated and directly admitted to the intensive care units (ICUs) when available. One small retrospective study showed lower mortality among sepsis patients enrolled than those not enrolled in the SFT (21% vs 43%) at the study hospital,[13] while another study did not find an association between SFT and mortality outcome.[14] These studies were subject to selection biases due to their retrospective nature.[13 14] Interventional studies to randomise patients to receive or not receive the SFT, however, would be unethical and impractical after the national recommendations have been implemented. It is increasingly recommended to evaluate the impact of healthcare interventions using routine data, particularly when a wide range of routinely collected data is available.[15]

Here, we analysed data from our prospective observational study of community-acquired sepsis patients presenting to a referral hospital in Thailand over 4 years (from March 2013 to January 2017)[16 17] to retrospectively evaluate the effectiveness of an SFT programme which was implemented at the study hospital in January 2015.

## MATERIAL AND METHODS
### Study design
We conducted a retrospective study to evaluate the effectiveness of the SFT programme by using the data of a prospective observational study (Ubon-sepsis).[16 17] The SFT programme was implemented at the study hospital in January 2015 until now as per national recommendations.[12] The SFT programme at the study hospital included (1) diagnostic criteria for attending physicians and medical teams to systematically identify sepsis patients on hospital admission (online supplemental table 1), (2) a recommended sepsis care protocol and (3) direct admission to the ICUs when available. The SFT programme at the study hospital was generated by the SFT committee of the study hospital (SB, SS, CB, PP, B-OS, OW, PC and PT) based on SSC 2012,[11] resource availability and local context.[12] The study hospital is a referral hospital to smaller district hospitals and provincial hospitals in three adjacent provinces. The referring hospitals were not involved in the SFT programme of the study hospital during the study period.

Details of the Ubon-sepsis cohort have been published elsewhere.[16 17] In short, the Ubon-sepsis research team, who were not attending physicians or medical teams at the study hospital, conducted a prospective observational study of community-acquired infections and sepsis from March 2013 to January 2017.[16 17] The research team prospectively enrolled adult patients ≥18 years old who were admitted to the general medical wards and medical ICUs with a primary diagnosis of infection made by the attending physician, were within 24 hours of admission to the study hospital and had three of 20 systemic manifestations of infection documented in the medical records (online supplemental table 2). The 20 systemic manifestations of the infections were consolidated from the 22 variables proposed as diagnostic criteria for sepsis for SSC 2012.[11] The study team sequentially screened all medical patients by reviewing admission logs in the emergency department (ED), medical wards and medical ICUs two times per day (morning and afternoon) on each working day. The Ubon-sepsis cohort was initiated in 2012 prior to the implementation of SFT at the study hospital. The research team was not involved in any clinical interventions; enrollment in the SFT programme and all medical treatment was performed by attending physicians and medical teams. The research team did not adjust the study protocol, inclusion criteria and exclusion criteria of the Ubon-sepsis cohort during the entire study period, and the research team recorded whether participants in the Ubon-sepsis cohort were enrolled in the SFT programme.

The reporting of this study follows the Strengthening the Reporting of Observational Studies in Epidemiology guidelines. Written, informed permission was obtained from participants prior to enrollment in the Ubon-sepsis cohort.

### Participants
For this study, we evaluated patients who were included into the Ubon-sepsis cohort and had community-acquired sepsis. Sepsis was defined as an infection with organ dysfunction in accordance with the 2016 international consensus (Sepsis-3) guidelines for sepsis.[1] Organ dysfunction was determined by a modified Sequential (sepsis-based) Organ Failure Assessment (SOFA) Score ≥2 as previously described.[16 17] The study was conducted in 2013 prior to the Sepsis-3 definition, and inotropic and vasopressor agent doses were not recorded into the case report form.[1 18] For the cardiovascular component of the SOFA Score, the scoring was modified such that subjects were scored a maximum of 2 (on a 4-point scale) if they received only dobutamine or dopamine, and scored a maximum of 3 if they received epinephrine or norepinephrine. For the respiratory component of the SOFA Score, as $PaO_2/FiO_2$ indices were not available for the majority of subjects due to infrequency of arterial blood gas tests, the score was modified as follows: subjects were scored a maximum of 2 (4-point scale) if they received advanced respiratory support (endotracheal tube, gas powered or electrical powered mechanical ventilation) and arterial blood gas test was not performed.[16 17] The Ubon-sepsis cohort excluded patients who were suspected of having hospital-acquired infections (determined by the attending physician), hospitalised within 30 days prior to

the current admission or hospitalised at any facility for a total duration longer than 72 hours prior to enrollment.

## Main exposure

Main exposure of the study was the SFT programme. All patients included in the Ubon-sepsis cohort from March 2013 to December 2014 who received standard care were considered as the non-exposed group. Patients included in the Ubon-sepsis cohort from January 2015 to January 2017 who received standard care or received care in the SFT programme by attending medical teams using their criteria on admission (online supplemental table 1) were considered as the additional non-exposed group or as the SFT group, respectively. The Ubon-sepsis research team were not involved in decision-making regarding enrollment to the SFT programme.

Patients in the non-exposed group received standard care according to local guidelines. Patients in the SFT group received the standard of care along with a recommended sepsis care protocol of the SFT programme. First, preprinted recommended doctor orders for the SFT programme were used as of January 2015 (online supplemental figure 1). The recommended orders included oxygen administration, intravenous fluid loading and fluid administration to achieve the recommended target of 30 mL/kg crystalloid, blood culture, recommended stat (immediate) doses and choices of parenteral antibiotics including ceftriaxone, ceftazidime, cloxacillin, metronidazole and gentamycin, contact ICU for ICU admission (if available), oxygen supplementation, close monitoring of vital signs and urine output, and a set of diagnostic tests including chest radiography, ECG, rapid blood glucose test, serum lactate, complete blood count, blood urea nitrogen, creatinine, electrolytes, liver function tests, albumin level, prothrombin time and partial thromboplastin time. Second, as of March 2016, the resuscitation workflow to normalise and maintain a mean arterial pressure ≥65 mm Hg, systolic blood pressure ≥90 mm Hg and urine output ≥0.5 mL/kg/hour within the first 6 hours was formally implemented and recommended (online supplemental figure 2). The resuscitation workflow included fluid resuscitation, measurement of central venous pressure and central venous oxygen saturation, administration of adrenergic agents, blood transfusion for haematocrit <30% and hydrocortisone if adequate fluid resuscitation and vasopressor therapy could not restore haemodynamic stability. The resuscitation workflow was pre-printed and included in the clinical chart of every SFT patient (together with pre-printed doctor's orders), and was recommended even if patients could not be admitted directly to the ICU. A separate set of documents, recommended management and recommended frequency of vital signs monitoring for nurses (ie, nurse notes for SFT patients) were also used for every SFT patient. Preparation and regular meetings to implement and monitor the SFT programme were organised by the SFT committee of Sunpasitthiprasong Hospital.

## Outcome measures

The primary outcome measure was 28-day mortality as recorded in the Ubon-sepsis cohort.[16] 28-Day mortality data were collected via telephone contact if subjects were no longer hospitalised and had been discharged alive.[16] The secondary outcome measures were sepsis management interventions; including antibiotics administration, blood cultures, mechanical ventilation, adrenergic agents, acute haemodialysis and placement of a urinary catheter within the first day of hospitalisation.[16 17]

## Sample size

The sample size of the study was determined by the sample size of Ubon-sepsis cohort. We assumed that about 50% of 3806 sepsis patients in the Ubon-sepsis cohort were enrolled after the implementation of the SFT programme, of which 50% were enrolled in the SFT programme (ie, 952 and 2854 patients were estimated to be the SFT and non-exposed group, respectively). We assumed that the mortality of the non-exposed group was 21% based on published data.[16 17] Our current sample size of 3806 would provide a power of 80% at an alpha error of 5% to detect a 4% difference in the mortality outcome.

## Statistical analysis

All sepsis patients were included in the analysis regardless of whether they were enrolled before or after the implementation of the SFT programme. We used the $\chi^2$ test and Mann-Whitney U test to compare the proportions of binary variables and median of continuous variables between groups, respectively. The IQR is presented as 25th and 75th percentiles.

In the primary analysis, we used multivariable Cox proportional hazard models to evaluate the effectiveness of SFT programme on 28-day mortality. The multivariable Cox proportional hazard model was used to adjust the difference between those receiving the SFT programme and the others.[19] To reduce bias in the model development, we used the previous multivariable Cox proportional hazard model as the base model,[16] added the SFT group variable and direct admission to the ICU, and modified by adding a time variable to represent possible changes over time and by using continuous modified SOFA Score on admission rather than as a binary variable (modified SOFA Score ≥2). Twenty eight patients enrolled in early 2017 were considered as enrolled in 2016. The continuous modified SOFA Score was used to improve regression adjustment for disease severity of the model. The other variables included in the model were gender, age group, transfer from other hospital, comorbidities (diabetes mellitus, chronic kidney disease, liver disease and malignancy) and blood culture positive for pathogenic organisms. We calculated the unadjusted and adjusted probability of survival at each timepoint using the Kaplan-Meier method (using the sts graph and stcurve command in STATA, respectively)

Using a conceptual framework, we also consider that admission directly to the ICU could also be a mediator between the SFT and the primary outcome; therefore, we developed another multivariable model not including the variable for direct admission to the ICU. The goodness of fit for the multivariable Cox proportional hazard model was tested with a Hosmer and Lemeshow test. For the Cox proportional hazard model, we assessed whether the HR was constant over time using Schoenfeld residuals.

For the secondary endpoints, we used multivariable logistic regression models with similar independent variables as the model for 28-mortality outcome and used each sepsis management process as an outcome. We estimated the total effect of the SFT on each sepsis management by using the multivariable logistic regression models adjusted for difference in characteristics and disease severity of the patients. This was because each sepsis management could be caused by characteristics of the patients, disease severity and the SFT.[20]

We also performed sensitivity analyses by using multivariable logistic regression model, excluding patients enrolled prior to the implementation of the SFT programme, and by replacing direct admission to the ICUs with admission to the ICUs within the first hospital day. All analyses were performed with STATA V.15.1 (StataCorp).

### Patient and public involvement

No patients were involved in setting the research question or the outcome measures, nor were they involved in developing plans for recruitment, design or implementation of the study. No patients were asked to advise on interpretation or writing the results. The results will be disseminated to the public through online social media.

## RESULTS

### Baseline characteristics

The observational cohort study (Ubon-sepsis) included 5001 patients presenting with community-acquired infections from March 2013 to January 2017, and 12 patients were excluded due to unknown 28-day mortality outcome. Three thousand eight hundred and six patients (76%) met criteria for sepsis within the first 24 hours of admission with a modified SOFA Score ≥2, and were included for the analysis. Figure 1 shows the flow of participants through the study. Among 3806 sepsis patients, 903 were enrolled in the SFT programme and considered as the SFT group, and 2903 were not enrolled in the SFT programme, received standard of care and considered as the non-exposed group. Of 2903 sepsis patients in the non-exposed group, 1636 were included in the observational cohort study prior to the implementation of SFT programme and 1267 were after the implementation of the programme.

Table 1 shows the characteristics of the study patients. Patients in the SFT group were older and more likely to have underlying diseases of diabetes mellitus, cerebrovascular diseases and dyslipidaemia. Patients included in the SFT group had higher severity of organ dysfunction determined by the modified SOFA Score compared with the non-exposed group (median 6 (IQR 4–9) vs 4 (IQR 3–6), p<0.001). A higher proportion of patients in the SFT group were admitted directly to the ICU compared with the non-exposed group (19% vs 5%, p<0.001).

### Primary outcomes

The primary outcome, mortality within 28 days, occurred in 205 of 903 (23%) in the SFT group and 574 of 2903 (20%) in the non-exposed group (figure 2A). In the primary analysis, patients in the SFT group were more likely to survive adjusted for baseline characteristics,

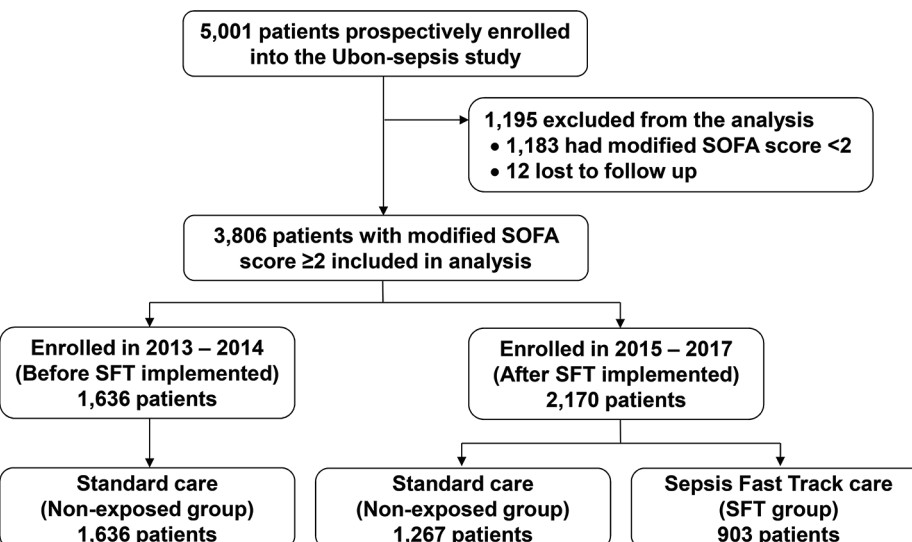

**Figure 1** Flow of participants through study. This study used the data of an observational study on sepsis patients (Ubon-sepsis) from March 2013 to January 2017 to evaluate the effectiveness of a Sepsis Fast Track (SFT) programme implemented at the study hospital in January 2015. SOFA, Sequential Organ Failure Assessment.

| Table 1 Baseline characteristics of sepsis patients enrolled in the Sepsis Fast Track programme* (SFT group) or standard of care (non-exposed group) | | |
|---|---|---|
| Characteristics | SFT group† (n=903) | Non-exposed group‡ (n=2903) |
| Male gender | 526 (58) | 1653 (57) |
| Age (years) (median (IQR)) | 63 (49–74) | 56 (39–70) |
| Age group (years) | | |
| 18–40 | 100 (11) | 647 (22) |
| >40–60 | 277 (31) | 875 (30) |
| >60–70 | 214 (24) | 513 (18) |
| >70 | 312 (35) | 868 (30) |
| Comorbidities | | |
| Hypertension | 239 (26) | 726 (25) |
| Diabetes mellitus | 213 (24) | 594 (20) |
| Chronic kidney disease | 129 (14) | 391 (13) |
| Dyslipidaemia | 66 (7) | 152 (5) |
| Heart disease | 48 (5) | 183 (6) |
| Lung disease | 67 (7) | 239 (8) |
| Liver disease | 33 (4) | 91 (3) |
| Cerebrovascular disease | 29 (3) | 55 (2) |
| Malignancy | 13 (1) | 47 (2) |
| HIV | 6 (1) | 33 (1) |
| Organ dysfunction | | |
| Modified SOFA Score (median (IQR)) | 6 (4–9) | 4 (3–6) |
| Renal dysfunction§ | 706 (78) | 1846 (64) |
| Cardiovascular dysfunction§ | 811 (90) | 1532 (53) |
| Coagulation dysfunction§ | 419 (46) | 1562 (54) |
| Liver dysfunction§ | 311 (34) | 822 (28) |
| Respiratory dysfunction§ | 337 (37) | 853 (29) |
| Central nervous system dysfunction§ | 166 (18) | 530 (18) |
| Transferred from other hospitals | 874 (97) | 2372 (84) |
| Duration of symptoms (median (IQR)) | 2 (1–3) | 3 (1–5) |
| ≤2 days | 505 (56) | 1191 (41) |
| 3–7 days | 362 (40) | 1488 (51) |
| >7 days | 36 (4) | 224 (8) |
| Presenting clinical syndromes¶ | | |
| Septic shock | 687 (76) | 733 (25) |
| Acute febrile illness | 206 (23) | 940 (32) |
| Lower respiratory infection | 223 (25) | 890 (31) |
| Sepsis | 225 (25) | 273 (9) |
| Others | 13 (1) | 456 (16) |
| Diarrheal illness | 150 (17) | 264 (9) |
| Direct admission to the ICU | 170 (19) | 128 (4) |
| Admission to the ICU within 24 hours of admission | 270 (29) | 370 (13) |
| Blood culture positive for pathogenic organisms | 175 (19) | 347 (12) |

Continued

| Table 1 Continued | | |
|---|---|---|
| Characteristics | SFT group† (n=903) | Non-exposed group‡ (n=2903) |
| Year | | |
| 2013 | N/A | 1047 (26) |
| 2014 | N/A | 1156 (29) |
| 2015 | 369 (39) | 956 (24) |
| 2016 | 556 (59) | 869 (21) |
| 2017 | 14 (1) | 22 (1) |

Values are number (%) unless stated otherwise.
*SFT programme was implemented at the study hospital in January 2015.
†Nine hundred and three patients of the Ubon-sepsis cohort were enrolled in SFT programme after the implementation of the SFT programme (figure 1).
‡Included 1636 and 1267 patients in the Ubon-sepsis cohort before and after the implementation of the SFT programme, respectively.
§Organ dysfunction defined as modified SOFA Score was ≥1 for each organ system.[16]
¶Patients may have more than one presenting clinical syndrome.
ICU, intensive care unit; SOFA, Sequential Organ Failure Assessment.

severity of sepsis and direct admission the ICUs (adjusted HR (aHR) 0.72, 95% CI 0.58 to 0.88, p=0.001, figure 2B and online supplemental table 3). Older age, higher modified SOFA Score, underlying disease of malignancy and chronic kidney disease, blood culture positive for pathogenic organisms and direct admission to the ICUs were associated with risk of mortality.

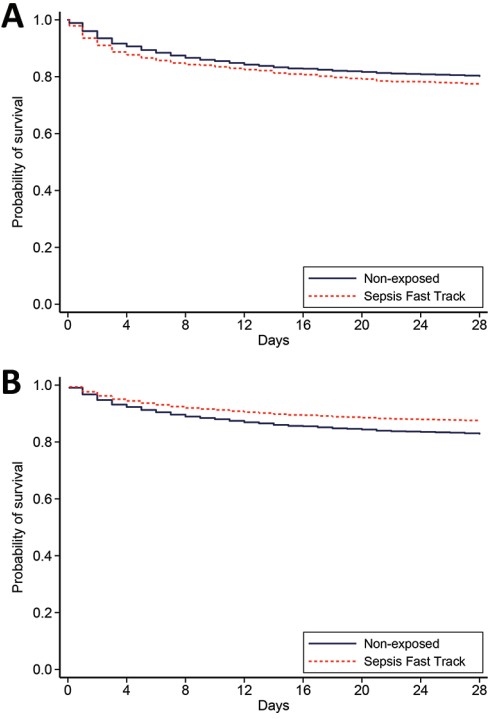

**Figure 2** (A) Unadjusted probability of survival and (B) adjusted probability of survival based on the multivariable Cox proportional hazard regression model.

## Sensitivity analyses

As we considered that direct admission to the ICU could be a mediator between the SFT and the outcome, a sensitivity analysis was performed by excluding the variable direct admission to the ICU (online supplemental table 4). The effect of SFT (aHR 0.77, 95% CI 0.63 to 0.94, p<0.001) was also observed. We observed that constant proportional hazard assumption was not strongly hold in one variable (the modified SOFA Score); therefore, additional sensitivity analyses were performed by using logistic multivariable models. The similar effect of SFT was observed (online supplemental tables 5 and 6).

We also performed a sensitivity analysis by excluding the 1636 patients enrolled in the observational study prior to the implementation of SFT programme. Similar differences in baseline characteristics were observed when comparing 903 patients in the SFT group to the 1267 patients in the non-exposed group enrolled after the implementation of the SFT programme (online supplemental table 7). A higher chance of survival in the SFT group compared with the non-exposed group was also observed (aHR 0.68, 95% CI 0.55 to 0.84, p<0.001; online supplemental table 7). We also performed another sensitivity analysis by replacing direct admission to the ICUs with admission to the ICUs within the first hospital day. Of 3806 patients, 640 (17%) were admitted to the ICUs within the first day of admission. A higher chance of survival in the SFT group compared with the non-exposed group was also observed (aHR 0.72, 95% CI 0.59 to 0.88, p=0.002).

## Secondary outcomes

Using multivariable logistic regression models, we found that patients in the SFT group were more likely to receive most sepsis management interventions than patients in the non-exposed group adjusting for baseline characteristics, severity of sepsis and direct admission to the ICU (table 2). Those included antibiotics, blood cultures, adrenergic agents and placement of a urinary catheter within the first day of hospitalisation. However, sepsis patients in the SFT group were less likely to receive

mechanical ventilation compared with those in the non-exposed group adjusting for baseline characteristics, severity of sepsis and direct admission to the ICUs group (adjusted OR (aOR) 0.30, 95% CI 0.24 to 0.38). We found that direct admission to the ICUs (aOR 5.77, 95% CI 4.20 to 7.92) and transfer from other hospitals (aOR 3.45, 95% CI 2.42 to 4.91) were strongly associated with the requirement of mechanical ventilation.

## DISCUSSION

In this study evaluating patients with community-acquired sepsis, enrollment into a programme to identify and initiate sepsis care implemented at the study hospital (SFT programme) was associated with 28% lower risk of mortality. In recent years, there has been an increasing need to understand benefit and cost effectiveness of implementation of sepsis care interventions in LMICs[11] because of concerns that international sepsis guidelines[11] may not be extrapolated to patients with tropical infectious diseases[7–9] and to resource-limited settings with poor ICU capacity.[10] In this study, we show the effectiveness of sepsis protocol modified based on resource availability in a tropical country, where causes of community-acquired sepsis include malaria and tropical viral diseases.[16 21 22] Majority of sepsis patients in our study were managed on the general wards, including those with respiratory failure or shock. Nonetheless, our study shows that enhancing sepsis care in the ED and general medical wards, as well as improving access to ICUs can reduce sepsis mortality in an LMIC.

The lower odds of receiving mechanical ventilation in the SFT group could be a sign of improved sepsis care. Patients in the SFT group are monitored closely either in or outside the ICUs, and the attending physicians aim to obviate the need for airway intubation when possible.[7] Attending physicians may tend to provide mechanical ventilation to patients in the non-exposed group based on broad indications such as (1) airway protection, (2) hypercapnic respiratory failure, (3) hypoxemic

| Table 2 | Clinical management within the first day of hospitalization | | | |
|---|---|---|---|---|
| **Clinical management*** | **SFT group (n=903)** | **Control group (n=2903)** | **Adjusted OR (95% CI)** | **P value** |
| Antibiotic | 897 (99%) | 2497 (86%) | 14.69 (6.36 to 33.91) | <0.001 |
| Blood culture | 829 (92%) | 2387 (82%) | 1.82 (1.35 to 2.45) | <0.001 |
| Urinary catheterisation | 862 (95%) | 1642 (57%) | 12.02 (8.41 to 17.20) | <0.001 |
| Acute dialysis | 10 (1.1%) | 23 (0.8%) | 1.96 (0.66 to 5.87) | 0.23 |
| Adrenergic agent | 706 (78%) | 902 (31%) | 11.53 (9.10 to 14.61) | <0.001 |
| Mechanical ventilation | 290 (32%) | 840 (29%) | 0.39 (0.31 to 0.49) | <0.001 |
| Direct admission to the ICU | 170 (18.8%) | 128 (4.4%) | 4.34 (2.96 to 6.36) | <0.001 |

*The effect of SFT on each clinical management was estimated by using the multivariable logistic regression models adjusted for admission year, gender, age, comorbidities, modified SOFA Score, transfer from other hospital, blood culture positive for pathogenic organisms and direct admission to the ICU.
ICU, intensive care unit; SFT, Sepsis Fast Track; SOFA, Sequential Organ Failure Assessment.

respiratory failure or (4) circulatory failure[23 24] because they may not be able to monitor patients' breathing and oxygen saturation as often as those enrolled in the SFT programme.

It is not surprising that patients in the SFT group had more organ dysfunction than those in the non-exposed group. This is because the severity of organ dysfunction among patients with septic shock, respiratory failure and alteration of conscious can be assessed clinically on admission, and those patients could be enrolled in the SFT programme when the laboratory test results were not yet available. However, the non-exposed group was defined as having sepsis based on clinical findings and all laboratory test results within 24 hours of admission (per protocol of Ubon-sepsis cohort study.[16 17] Therefore, the non-exposed group could use laboratory test results (ie, liver function tests, creatinine level, international normalised ratio and activated partial thromplastin time) from blood specimens drawn on admission. Therefore, the SFT programme was more likely to enrol patients with obvious signs of sepsis and septic shock; such as acute respiratory failure and hypotension, while Ubon-sepsis cohort could include sepsis patients with relatively lower modified SOFA Scores.

## Comparison with other studies

Our study is not the first to evaluate effectiveness of sepsis intervention in LMICs. Early recognition and protocol directed intervention improves outcomes of sepsis in adults[25–27] and severe infection in children[28] in LMICs. The optimal method of fluid resuscitation in sepsis in tropical LMICs has not been determined.[8 25 29 30] Our resuscitation protocol is a simple guideline, and the SFT recommend doctors to be careful and adjust fluid resuscitation based on preliminary diagnoses, underlying diseases and rapid diagnostic test results (ie, if sepsis is caused by malaria or dengue infection). The implementation of the SFT programme in our study hospital and in Thailand is consistent with the recommendation of 'SCAN-TEACH-TREAT' programme developed by Sepsis in Resource-Limited Settings Workgroup of the Surviving Sepsis Campaign.[7] The SFT programme evaluated resources in the setting (SCAN component), focused on educational interventions on early recognition and management of sepsis among medical personnel including physicians, nurses and students (TEACH component) and implemented pragmatic and simple bundles into practice (TREAT component). In addition, the SFT programme has the strong support and endorsement of local health and governmental leaders.[12]

## Strength and limitations of the study

This study features four strengths. First, the study hospital used the published framework, SCAN-TEACH-TREAT programme to develop a context-specific quality of care improvement for sepsis,[7] and we closely monitor and evaluate the effectiveness of an intervention. Second, the study took advantage of a robust prospective observational study design that strengthened causal inference by providing pre-intervention information, having an appropriate non-exposed group from both pre-intervention and post-intervention periods, and controlling important confounding factors (ie, the modified SOFA Score) which were measured systematically throughout the study period. Third, this study incorporated several predictors of interest (measured sepsis management interventions and admission to the ICUs). This allows us to identify that the increase in most measured sepsis interventions associated with the SFT programme and that led to the survival benefit among sepsis patients. Fourth, the focus on sepsis at a public tertiary-care hospital in Thailand helped us to estimate the effect of sepsis protocol in a tropical resource-limited setting with large sample size.

Our study had several limitations. First, a modified SOFA Score was used because the dosage of dobutamine, dopamine, epinephrine and norepinephrine were not recorded and arterial blood gases were rarely performed. The modified SOFA Score (maximum 23) may be lower than the SOFA Score (maximum 24). Nonetheless, the modified SOFA Score is strongly associated with mortality in sepsis.[16 17] Second, the proportional hazards assumption was met for all variables, including the main variable (the SFT), except one controlled variable (the modified SOFA Score). The adjusted effect estimates could be under or overestimated due to residual confounding factors such as improvement of care and profile of organ failure recognition overtimes. Third, due to the use of observational data, the observed effects of the SFT on 28-day mortality in our study should be interpreted conservatively as an association rather than a causation.

## Conclusions and future implications

Our study successfully demonstrated effectiveness of a sepsis programme implemented in an LMIC. Measuring effectiveness of a sepsis programme is a complex issue, and we used a data of a prospective observational study and carefully controlled for severity of sepsis and temporal trends in our analyses. Care in sepsis patients improved after the implementation of the programme. Additional research is needed to better understand cost of the intervention, long-term benefits and impact of the programme on a national scale. National strategies aimed at saving lives from sepsis in LMICs should be encouraged. Such strategies should include analysis of resources and local circumstances, followed by development, implementation and assessment of customised programmes.

**Author affiliations**
[1]Department of Internal Medicine, Sunpasitthiprasong Hospital, Ubon Ratchathani, Thailand
[2]Mahidol Oxford Tropical Medicine Research Unit, Faculty of Tropical Medicine, Mahidol University, Bangkok, Thailand
[3]Emergency Department, Sunpasitthiprasong Hospital, Ubon Ratchathani, Thailand
[4]Department for Research Support and Development, Sunpasitthiprasong Hospital, Ubon Ratchathani, Thailand
[5]Department of Internal Medicine, Buddhachinaraj Phitsanulok Hospital, Phitsanulok, Thailand

[6]Department of Critical Care Medicine, University of Pittsburgh, Pittsburgh, Pennsylvania, USA

[7]Division of Pulmonary, Critical Care and Sleep Medicine, University of Washington, Seattle, Washington, USA

[8]Centre for Tropical Medicine and Global Health, Nuffield Department of Medicine, University of Oxford, Oxford, UK

[9]Department of Microbiology and Immunology, Faculty of Tropical Medicine, Mahidol University, Bangkok, Thailand

[10]Department of Tropical Hygiene, Faculty of Tropical Medicine, Mahidol University, Bangkok, Thailand

**Acknowledgements** We thank all patients, their relatives and staff of the Sunpasitthiprasong hospital who participated in the study. We thank Mayura Malasit, Praweennuch Watanachaiprasert, Chayamon Krainoonsing, Passaraporn Kesaphun, Nannicha Jirapornuwat, Gumphol Wongsuvan, Areeya Faosap, Yaowaret Dokket, Sukhumal Pewlaorng, Jintana Suwannapruek, Prapass Wannapinij and Diane Tomita for their clinical, laboratory and administrative support.

**Contributors** NPJD, TEW and DL obtained grant funding. SB, VH, PT, TEW and DL contributed to study conception, development and study design. SB, VH, PT, TEW and DL contributed to study conduct, data collection and study administration. VH and DL performed the statistical analysis and interpreted the data and had full access to all of the data in the study. Both authors can take responsibility for the integrity of the data and the accuracy of the data analysis. DL is a guarantor. SB, VH, PT, TEW and DL wrote the first draft of a manuscript, with input from SS, CB, PC, KR and AMD. PP, B-OS, OW and RC provided scientific or administrative support. All authors contributed to results interpretation, critically revised and approved the final submitted manuscript. The corresponding author attests that all listed authors meet authorship criteria and that no others meeting the criteria have been omitted.

**Funding** The study was funded by the Wellcome Trust (090219/Z/09/Z) and National Heart, Lung and Blood Institute, National Institutes of Health (R01HL113382). DL is supported by an intermediate fellowship from the Wellcome Trust (101103/Z/13/Z).

**Disclaimer** The funders had no role in the design and conduct of the study, all study procedures, data collection, data analyses, data interpretation, writing of the report and the decision to submit the article for publication.

**Competing interests** None declared.

**Patient consent for publication** Not required.

**Ethics approval** The study was conducted in full compliance with the principles of good clinical practice (GCP), and the ethical principles of the Declaration of Helsinki. The study protocol and related documents were approved by Sunpasitthiprasong Hospital Ethics Committee (039/2556), the Ethics Committee of the Faculty of Tropical Medicine, Mahidol University (MUTM2012-024-01), the University of Washington Institutional Review Board (42988) and the Oxford Tropical Research Ethics Committee at the University of Oxford (OXTREC172-12). Signed or fingerprinted informed consent was obtained from the participants or their representatives before enrollment.

**Provenance and peer review** Not commissioned; externally peer reviewed.

**Data availability statement** Data are available in a public, open access repository. The final database with the data dictionary are publicly available online https://doi.org/10.6084/m9.figshare.12102627.

**ORCID iDs**
Viriya Hantrakun http://orcid.org/0000-0002-6727-3404
Direk Limmathurotsakul http://orcid.org/0000-0001-7240-5320

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
