## [Reviewer comments · BMJ Open]

ARTICLE DETAILS

TITLE (PROVISIONAL)	Effectiveness of a sepsis programme in a resource-limited setting: a retrospective analysis of data of a prospective observational study (Ubon-sepsis)
AUTHORS	Booraphun, Suchart; Hantrakun, Viriya; Siriboon, Suwatthiya; Boonsri, Chaiyaporn; Poomthong, Pulyamon; Singkaew, Bung-Orn; Wasombat, Oratai; Chamnan, Parinya; Champunot, Ratapum; Rudd, Kristina; Day, Nicholas; Dondorp, Arjen; Teparrukkul, Prapit; West, Timothy Eoin; Limmathurotsakul, Direk

VERSION 1 – REVIEW

REVIEWER	Bruno Adler Maccagnan Pinheiro Besen Medical ICU, Internal Medicine Department, Hospital das Clínicas HCFMUSP, Faculdade de Medicina, Universidade de São Paulo, São Paulo (SP), Brasil
REVIEW RETURNED	30-Jun-2020

GENERAL COMMENTS	Dear authors, Thank you for the opportunity to review your manuscript. Overall manuscript description: The authors designed a quasi-experimental study to evaluate the effectiveness of a "Sepsis fast track" (SFT) program implemented in Thailand for the management of suspected patients with sepsis and septic shock in a single hospital. As I could understand, the hospital only receives patients referenced from other institutions. The SFT was implemented in Thailand from January, 2015. The SFT involved early recognition, deployment of treatment strategies according to international guidelines and a recommendation to transfer the patient to the ICU if the bed was available. Although the program started in January, 2015, it was at the clinician discretion to start the SFT or not, which is a major issue in the interpretation of this manuscript. Patient data was collected from a prospective registry (Ubon-sepsis cohort) and data from the SFT was collected separately. The main research question was whether the SFT was effective in reducing 28-day mortality from sepsis. The authors also evaluated the impact of ICU admission in the outcome through interaction tests. The main results were: (1) after multivariable adjustment in a Cox model, the SFT was associated with a lower risk of death (aHR 0.70); and (2) being admitted to the ICU was not a effect modifier ($p = 0.71$). While the authors should be congratulated for their efforts in evaluating this intervention, there are some major issues that need be addressed. Some of them are inter-related, but for the sake of clarity, I will divide them in different topics.
--

	Major issues: (1) The authors state this is a natural experimental study and present it with CONSORT guidelines. While the implementation of the intervention in January, 2015 could be viewed as a quasi-experiment (or natural experiment, it's hard to separate these two interrelated concepts), the data analysis relied on patients actually receiving the intervention from the clinicians discretion. Therefore, in this reviewer's opinion, this is an observational study and should be presented as per STROBE guidance. If a natural experiment was to be described (as the introduction suggests that this was an impact evaluation), the exposure should have been being admitted to the hospital once the SFT was initiated and not the clinician use of the SFT at his/her discretion. This is a major issue in your design and analysis. Impact evaluation of the programme (was it effective in being implemented considering process measures and did this lead to improved outcomes) is different from having benefited from the SFT protocol at the clinicians discretion. (2) The authors developed and conducted a causal question from observational data. This leads to some important directions: first, presenting a causal directed acyclic graph would be interesting or at least the minimum set of confounders to address this specific causal question. Second, the authors present a "Table 2" consistent with the "Table 2 fallacy" (American Journal of Epidemiology, Volume 177, Issue 4, 15 February 2013, Pages 292–298) which need not to be presented in this kind of study. Adjustments should be done based on the causal assumptions and all variables included, except if degrees of freedom do not allow for such model to converge properly - a limitation would need to be acknowledged in this scenario. Third, presenting p-values for the Table 1 is not necessary (they are not actually tested hypotheses), although this is still required by many journals and should be considered on a case-by-case basis. Regarding the selection of confounders, I missed performance status and palliative care status and also a description of chronic organ failure more explicit. (3) If we consider the SFT use by the clinician as the exposure, including admission to the ICU or other process measures in a model for adjustment is not correct, since you condition on a mediator of the outcome. The analysis of ICU admission as a confounder or effect modifier (using interaction tests), therefore, does not seem to be correct, in this reviewer's opinion. (4) Issues of writing and clarity: overall, the manuscript is well written, but it can be reduced substantially. As suggestions:  - The introduction is too long (aim for a one page Introduction if possible). It should be focused on the literature gap regarding the intervention, not on the method. Allow the method to be better explained / described in the methods session. - In the methods session, the concept and rationale for a natural experiment is explained too many times. Try to explain it only once and in a cleaner way (but address the issues of the first comments of this reviewer). - Try to separate at least conceptually process measures (outcome measures related directly with the SFT) from other clinical outcomes, such as the need for mechanical ventilation or dialysis.
--	---

	- The second and third paragraphs of the discussion are a bit loose. Please consider rewriting them to address specific issues of the manuscript. (5) Sample size calculation: by reading the manuscript, it seems clear to me that there was no a priori sample size calculation. You certainly have a reasonable sample size that allows for inference and statistical adjustment. Instead of focusing on previous assumptions (which I believe were probably not actually done), I would suggest to focus on the sample size and that there would be enough power to detect a 4 or 5% difference in the primary outcome (28-day mortality). (6) Missing data issues: there is no mention to how missing data was handled. It's quite hard to believe that there was no missing data in this dataset. Please provide how this was assessed and tackled during the analysis stage of this study. (7) Some important references (that I know of) from LMICs are missing. I would suggest at least two: "Crit Care. 2017 Oct 31;21(1):268. doi: 10.1186/s13054-017-1858-z"; "Intensive Care Med. 2014 Feb;40(2):182-91. doi: 10.1007/s00134-013-3131-5" Minor issues: (1) Estimating the NNT from this study seems to be a too strong interpretation given the study design and limitations. (2) Avoid confusing mortality (analysed with logistic regression) with survival (analysed with Cox-regression). Example: page 12, line 220. (3) Please describe if and how the the proportional hazards assumption was evaluated. If so, did it hold? (4) Please avoid the use of the term study group assignment and use exposures instead. (5) Page 16, line 287: "there was a borderline evidence showing that 28-day mortality of the SFT group was higher than control group (23% vs 20%, p=0.06)". This is actually no evidence. I would suggest to avoid this interpretation of p-values, especially since this is an unadjusted estimate. (6) Page 14, line 241: "performed" is probably a typo. (7) I would suggest to present a survival plot, given that survival analysis was the primary method of data analysis. Again, I would like to thank for the opportunity to review your manuscript. Kind regards, Bruno Besen
--	---

REVIEWER	Arthur Kwizera Makerere University University College of Health Sciences
REVIEW RETURNED	05-Sep-2020

GENERAL COMMENTS	The authors present a quality improvement study that demonstrates improvement in sepsis related survival outcomes in a resource constrained country. They conduct the study using a natural experiment, a rarely used interventional study method in sepsis studies. I would recommend publication of this after a some revisions as follows.  1. This study comes out as being a retrospective analysis of patient data. If this is the case, this must be clearly stated in the abstract and methods section. If not, then my confusion would explain that more clarity is needed. 2. How long was the SFT implementation for? Did it just involve introduction of a doctors orders form? was training done? Were other 3. I see two phases, the second of which involved addition of a fluid resuscitation protocol. Why did it take a year to introduce it? How many patients were recruited before the fluid resuscitation protocol? May this have had an effect on outcomes? It would be good to perform an analysis of the group recruited after the fluid protocol. 4. There were statistically significant differences of interest in the baseline comparisons. Were referring hospitals involved in the SFT program 5. In the discussion, the limitation of generalisability is not a true limitation. The strength of the study is that it utilised an already published framework (Scan-Teach-Treat) to develop a context specific quality of care improvement program for sepsis. 6. At the start of the discussion, I would rewrite the opening statement without so many figures since you have already shown this in the results section. for example In this natural experiment evaluating patients with community acquired sepsis, enrollment into a programme to identify and initiate sepsis care implemented at the study hospital was associated with a 30% reduction in mortality. (Thats the big message) It is worthwhile to note that the study sample size calculation was well powered to answer the primary research questions. This does not always happen in sepsis research. I congratulate the investigators.
---

VERSION 1 – AUTHOR RESPONSE

Reviewer: 1 Dr Besen

Overall manuscript description:

The authors designed a quasi-experimental study to evaluate the effectiveness of a "Sepsis fast track" (SFT) program implemented in Thailand for the management of suspected patients with sepsis and septic shock in a single hospital. As I could understand, the hospital only receives patients referenced from other institutions. The SFT was implemented in Thailand from January, 2015. The SFT involved early recognition, deployment of treatment strategies according to international guidelines and a recommendation to transfer the patient to the ICU if the bed was available. Although the program started in January, 2015, it was at the clinician discretion to start the SFT or not, which is a major issue in the interpretation of this manuscript. Patient data was collected from a prospective registry (Ubon-sepsis cohort) and data from the SFT was collected separately. The main research question was whether the SFT was effective in reducing 28-day mortality from sepsis. The authors also

evaluated the impact of ICU admission in the outcome through interaction tests. The main results were: (1) after multivariable adjustment in a Cox model, the SFT was associated with a lower risk of death (aHR 0.70); and (2) being admitted to the ICU was not an effect modifier ($p = 0.71$). While the authors should be congratulated for their efforts in evaluating this intervention, there are some major issues that need to be addressed. Some of them are inter-related, but for the sake of clarity, I will divide them in different topics.

Response: We are grateful for the comments by the reviewer. We would like to note that the study hospital receives patients from both referral from other institutions and direct admission. This was shown in Table 1 that 97% and 84% of patients in the SFT group and the control group were transferred from other hospitals.

Major issues:

(1) The authors state this is a natural experimental study and present it with CONSORT guidelines. While the implementation of the intervention in January, 2015 could be viewed as a quasi-experiment (or natural experiment, it's hard to separate these two interrelated concepts), the data analysis relied on patients actually receiving the intervention from the clinicians' discretion. Therefore, in this reviewer's opinion, this is an observational study and should be presented as per STROBE guidance. If a natural experiment was to be described (as the introduction suggests that this was an impact evaluation), the exposure should have been being admitted to the hospital once the SFT was initiated and not the clinician use of the SFT at his/her discretion. This is a major issue in your design and analysis. Impact evaluation of the programme (was it effective in being implemented considering process measures and did this lead to improved outcomes) is different from having benefited from the SFT protocol at the clinicians' discretion.

Response: We are thankful for the advice and have revised the manuscript to be a prospective observational study. The title of the study has been revised as follows, "Effectiveness of a sepsis programme in a resource-limited setting". The term 'natural experiment' has been revised throughout the manuscript.

(2) The authors developed and conducted a causal question from observational data. This leads to some important directions: first, presenting a causal directed acyclic graph would be interesting or at least the minimum set of confounders to address this specific causal question. Second, the authors present a "Table 2" consistent with the "Table 2 fallacy" (American Journal of Epidemiology, Volume 177, Issue 4, 15 February 2013, Pages 292–298) which need not to be presented in this kind of study. Adjustments should be done based on the causal assumptions and all variables included, except if degrees of freedom do not allow for such model to converge properly - a limitation would need to be acknowledged in this scenario. Third, presenting p-values for the Table 1 is not necessary (they are not actually tested hypotheses), although this is still required by many journals and should be considered on a case-by-case basis. Regarding the selection of confounders, I missed performance status and palliative care status and also a description of chronic organ failure more explicit.

Response: We are thankful for the advice. To reduce the possibility of readers to misinterpret the causal assumption of the manuscript, Table 3 has been moved to be Supplementary Table 4. As we changed the concept of study design used to be a prospective observational study, we did not use causal assumptions at the level of natural experiment design.

Regarding the presentation of p-values in tables, we believe that readers, including clinicians, would be interested in seeing the p values for the Table 1 and Table 2 as presented. Therefore, we decided to keep the p values for the Table 1 and Table 2, and additionally included the following lines in the footnote as suggested by the reference for clarity, "The hazard ratio of SFT was adjusted for gender, age group, transferred from other hospital, modified SOFA score, comorbidities, blood culture positive for pathogenic organisms, year and direct admission to the ICU. The hazard ratios of other variables could be considered as the controlled direct effect of those variables." We could make further

changes including removing p value for the Table 1 and removing Table 2 upon editor's request.

(3) If we consider the SFT use by the clinician as the exposure, including admission to the ICU or other process measures in a model for adjustment is not correct, since you condition on a mediator of the outcome. The analysis of ICU admission as a confounder or effect modifier (using interaction tests), therefore, does not seem to be correct, in this reviewer's opinion.

Response: We are thankful for the advice, and can understand the conceptual framework caused by SFT to direct admission to the ICU. We additionally provided a different model without direct admission to the ICU as Supplementary Table 3. The following paragraphs have been added into the method and result section, "Using a conceptual framework, we also consider that admission directly to the ICU could also be caused by the SFT; therefore, we developed another multivariable model not including the variable for direct admission to the ICU.)" and "Using a conceptual framework, we considered that admission directly to the ICU could also be caused by the SFT; therefore, we developed another multivariable model that did not adjust for direct admission to the ICU. We observed that the effect of the SFT was comparable (aHR 0.77, 95% CI 0.63 to 0.94; Supplementary Table 3).", respectively.

(4) Issues of writing and clarity: overall, the manuscript is well written, but it can be reduced substantially. As suggestions:

- The introduction is too long (aim for a one page Introduction if possible). It should be focused on the literature gap regarding the intervention, not on the method. Allow the method to be better explained / described in the methods session.

Response: The introduction has been revised accordingly.

- In the methods session, the concept and rationale for a natural experiment is explained too many times. Try to explain it only once and in a cleaner way (but address the issues of the first comments of this reviewer).

Response: The concept of the natural experiment has been removed.

- Try to separate at least conceptually process measures (outcome measures related directly with the SFT) from other clinical outcomes, such as the need for mechanical ventilation or dialysis.

Response: The manuscript has been revised for clarity.

- The second and third paragraphs of the discussion are a bit loose. Please consider rewriting them to address specific issues of the manuscript.

Response: The second and third paragraphs of the discussion have been revised. We considered that those phenomena are of interest to clinicians. We revised both paragraphs to be more specific to the issue of the SFT, discussing how patients were received or not received the secondary outcomes – specifically the mechanical ventilation (Paragraph 2) - and how patients were included or not included in the SFT in the real world setting (Paragraph 3)

(5) Sample size calculation: by reading the manuscript, it seems clear to me that there was no a priori sample size calculation. You certainly have a reasonable sample size that allows for inference and statistical adjustment. Instead of focusing on previous assumptions (which I believe were probably not actually done), I would suggest to focus on the sample size and that there would be enough power to detect a 4 or 5% difference in the primary outcome (28-day mortality).

Response: The sentence has been revised as suggested, "Our current sample size of 3,806 would provide a power of 80% at an alpha error of 5% to detect a 4% difference in the mortality outcome."

(6) Missing data issues: there is no mention to how missing data was handled. It's quite hard to believe that there was no missing data in this dataset. Please provide how this was assessed and

tackled during the analysis stage of this study.

Response: For the missing data of primary outcomes, we excluded them from the analyses. We have also revised a paragraph to explain how we handle missing data as follows, "For the cardiovascular component of the SOFA score, the scoring was modified such that subjects were scored a maximum of 2 (on a 4-point scale) if they received only dobutamine or dopamine, and scored a maximum of 3 if they received epinephrine or norepinephrine. For the respiratory component of the SOFA score, as PaO₂/FiO₂ indices were not available for the majority of subjects due to infrequency of arterial blood gas tests, the score was modified as follows: Subjects were scored a maximum of 2 (4-point scale) if they received advanced respiratory support (endotracheal tube, gas powered or electrical powered mechanical ventilation) and arterial blood gas test was not performed."

(7) Some important references (that I know of) from LMICs are missing. I would suggest at least two: "Crit Care. 2017 Oct 31;21(1):268. doi: 10.1186/s13054-017-1858-z"; "Intensive Care Med. 2014 Feb;40(2):182-91. doi: 10.1007/s00134-013-3131-5"

Response: The references have been added as suggested.

Minor issues:

(1) Estimating the NNT from this study seems to be a too strong interpretation given the study design and limitations.

Response: The section of NNT has been removed.

(2) Avoid confusing mortality (analysed with logistic regression) with survival (analysed with Cox-regression). Example: page 12, line 220.

Response: The manuscript has been revised as suggested.

(3) Please describe if and how the the proportional hazards assumption was evaluated. If so, did it hold?

Response: The following sentences in the method and discussion sections have been added for clarity, "For the Cox Proportional Hazard model, we assessed whether the hazard ratio was constant over time using Schoenfeld residuals." and "Second, the proportional hazards assumption was met for all variables, including the main variable (the SFT), except one controlled variable (the modified SOFA score). The adjusted effect estimates could be under or overestimated due to residual confounding factors, which inhered in the study design such as improvement of care and profile of organ failure recognition overtimes."

We explored multiple methods; including categorization and transformation of the variable modified SOFA score, but none improved the model. Therefore, we decided to keep the final model as presented based on clinical significance of modified SOFA score.

(4) Please avoid the use of the term study group assignment and use exposures instead.

Response: The term "study group assignment" has been removed as suggested. The term "non-exposed group" was used throughout the manuscript.

(5) Page 16, line 287: "there was a borderline evidence showing that 28-day mortality of the SFT group was higher than control group (23% vs 20%, p=0.06)". This is actually no evidence. I would suggest to avoid this interpretation of p-values, especially since this is an unadjusted estimate.

Response: The sentence has been revised as follows, "The 28-day mortality of the SFT group and the control group was 23% and 20%, respectively."

(6) Page 14, line 241: "preformed" is probably a typo.

Response: The typo has been corrected.

(7) I would suggest to present a survival plot, given that survival analysis was the primary method of data analysis.

Response: Due to the complexity of the study; including the difference in survival between groups in the models without and with adjustment, we decided not to present a survival plot.

Reviewer: 2 Dr Kwizera

They conduct the study using a natural experiment, a rarely used interventional study method in sepsis studies. I would recommend publication of this after a some revisions as follows.

1. This study comes out as being a retrospective analysis of patient data. If this is the case, this must be clearly stated in the abstract and methods section. If not, then my confusion would explain that more clarity is needed.

Response: The abstract has been revised for clarity as follows, "A retrospective analysis using the data of a prospective observational study on sepsis patients (Ubon-sepsis) from March 2013 to January 2017". The method has been revised as follows, "We conducted a retrospective study to evaluate the effectiveness of the SFT programme by using the data of a prospective observational study (Ubon-sepsis)."

2. How long was the SFT implementation for? Did it just involve introduction of a doctors orders form? was training done? Were other

Response: The SFT has been implemented until now as per national recommendations. The SFT involve doctors and nurses, and include trainings as described in the reference 12.

3. I see two phases, the second of which involved addition of a fluid resuscitation protocol. Why did it take a year to introduce it? How many patients were recruited before the fluid resuscitation protocol? May this have had an effect on outcomes? It would be good to perform an analysis of the group recruited after the fluid protocol.

Response: We took account of time by having a variable representing year in the model as shown in the Table 2. The total number of 1403 patients were observed from 2015 to 2017. We did not observe strong association between year and outcome adjusted for all other variables including exposure to the SFT programme. We would like to avoid subgroup analysis, we did not pre-plan for an interaction test between year and the SFT programme, and we considered that our study did not have enough power to evaluate additional interaction tests.

4. There were statistically significant differences of interest in the baseline comparisons. Were referring hospitals involved in the SFT program

Response: No, referring hospitals did not involve in the SFT programme during the study period.

5. In the discussion, the limitation of generalisability is not a true limitation. The strength of the study is that it utilised an already published framework (Scan-Teach-Treat) to develop a context specific quality of care improvement program for sepsis.

Response: The discussion section has been revised as follows, "This study features four strengths. First, the study hospital utilized the published framework, SCAN-TEACH-TREAT programme to develop a context specific quality of care improvement for sepsis, and we closely monitor and evaluate the effectiveness of the intervention."

6. At the start of the discussion, I would rewrite the opening statement without so many figures since you have already shown this in the results section.

Response: The sentence has been revised as suggested, "In this study evaluating patients with community-acquired sepsis, enrollment into a programme to identify and initiate sepsis care implemented at the study hospital (SFT programme) was associated with 28% lower risk of mortality."

It is worthwhile to note that the study sample size calculation was well powered to answer the primary research questions. This does not always happen in sepsis research. I congratulate the investigators. Response: We are grateful for the comments.

All contributing authors have reviewed and concurred with the revised manuscript.

VERSION 2 – REVIEW

REVIEWER	Bruno Besen Hospital das Clínicas, University of Sao Paulo Medical School
REVIEW RETURNED	13-Nov-2020

GENERAL COMMENTS	Dear authors, Thank you for the revision of the manuscript considering some of my suggestions. I still have some issues that are relevant to be addressed: 1) Please, include the study design in the Title of manuscript (i.e., retrospective cohort) 2) Page 8, line 162: "or were received care" looks like a typo. 3) In the evaluation of ICU admission, I am still worried this is a mediator. What do you mean by direct admission to the ICU? Patients were transferred from another hospital and admitted directly to the ICU? If so, then this is a baseline characteristic and can be considered a confounder. However, if it is what I thought it was the last time (an intervention of the sepsis fast track program after ED admission to admit the patient soon to the ICU), then it is a process measure that's actually a mediator. In this scenario, it cannot be interpreted with interaction tests, which are tests to evaluate effect modification, not mediation. Please revise properly to provide this information more clearly and to avoid the reader to misinterpret your findings. If it is a mediator, please refrain from presenting interaction tests and consider presenting the total, direct and indirect effects of the SFT accounting for admission to the ICU (an important mediator in your cohort given the low availability of resources). 4) While I agree with the authors that presenting p-values in the table 1 is still a matter of style (with which I personally disagree), presenting effect estimates from a multivariable model from a conceptual framework for statistical adjustment is conceptually wrong (i.e., Table 2 fallacy), since the estimates should not be interpreted, except for the exposures of interest. Instead, presenting a table 2 with sensitivity analyses for the exposure of interest would be more cleaner and informative to the reader regarding your primary research question. In this scenario, since you are publishing a survival analysis, presenting the number of events and follow-up time for both groups would also be informative, since you find presenting the Survival plots unnecessary.
---

	- Furthermore, I find the information from Supplementary Table 4 much more informative than the information presented in the Table 2, which is mostly non interpretable. 5) Given the nonproportional hazards in your model (not addressed with stratification, inclusion of time-varying covariates or other technique to allow for nonproportional hazards), I suggest you consider presenting results in a logistic regression framework either in a sensitivity analysis or as the primary statistical analysis method. By refraining the reader from observing a survival plot (both an unadjusted Kaplan-Meier curve and a plot using the stcurve command (stratified by SFT) after fitting the Cox model in Stata), I find the correct interpretation of results and applicability really frail. 6) Acknowledge in the limitations session that this is a retrospective cohort study still subject to confounding and that these results do not necessarily imply causation. While I have made these criticisms, I reinforce the dataset is strong and should be published, but the statistical analysis methods and interpretation seem to need some improvement. Thank you for the opportunity to review your manuscript.
--	--

VERSION 2 – AUTHOR RESPONSE

Reviewer: 1 Dr Besen

Thank you for the revision of the manuscript considering some of my suggestions.

I still have some issues that are relevant to be addressed:

1) Please, include the study design in the Title of manuscript (i.e., retrospective cohort)

Response: The title has been changed to “Effectiveness of a sepsis programme in a resource-limited setting: a retrospective analysis of data of a prospective observational study (Ubon-sepsis)”.

2) Page 8, line 162: "or were received care" looks like a typo.

Response: A correction on typo error in line 162 was made as follows:

“Patients included in the Ubon-sepsis cohort from January 2015 to January 2017 who received standard care or received care in the SFT programme by attending medical teams using their criteria on admission (Supplementary Table 1) were considered as the additional non-exposed group or as the SFT group, respectively.”

3) In the evaluation of ICU admission, I am still worried this is a mediator. What do you mean by direct admission to the ICU? Patients were transferred from another hospital and admitted directly to the ICU? If so, then this is a baseline characteristic and can be considered a confounder. However, if it is what I thought it was the last time (an intervention of the sepsis fast track program after ED admission to admit the patient soon to the ICU), then it is a process measure that's actually a mediator. In this scenario, it cannot be interpreted with interaction tests, which are tests to evaluate effect modification, not mediation. Please revise properly to provide this information more clearly and to avoid the reader to misinterpret your findings. If it is a mediator, please refrain from presenting interaction tests and consider presenting the total, direct and indirect effects of the SFT accounting for admission to the ICU (an important mediator in your cohort given the low availability of resources).

Response: Patients with direct admission to the ICU in our study were the patients admitted to the ICU upon admission day (both SFT and non-SFT patients). We agree that the direct admission to the ICU could be a mediator. We revised the text and analysis as follows, “As we considered that direct admission to the ICU could be a mediator between the SFT and the outcome, a sensitivity analysis was performed by excluding the variable direct admission to the ICU (Supplementary Table 4). The effect of SFT (aHR 0.77, 95% CI 0.63-0.94, $p < 0.001$) was also observed.”

4) While I agree with the authors that presenting p-values in the table 1 is still a matter of style (with which I personally disagree), presenting effect estimates from a multivariable model from a conceptual framework for statistical adjustment is conceptually wrong (i.e., Table 2 fallacy), since the estimates should not be interpreted, except for the exposures of interest. Instead, presenting a table 2 with sensitivity analyses for the exposure of interest would be more cleaner and informative to the reader regarding your primary research question. In this scenario, since you are publishing a survival analysis, presenting the number of events and follow-up time for both groups would also be informative, since you find presenting the Survival plots unnecessary.
- Furthermore, I find the information from Supplementary Table 4 much more informative than the information presented in the Table 2, which is mostly non interpretable.

Response: As suggested, we removed p-values in Table 1. We also removed Table 2 and presented the main exposure result as text in the result section. We agree with the reviewer that information in Supplementary Table 4 (clinical Management) is informative; therefore we moved and presented it in the main manuscript as Table 2.

5) Given the nonproportional hazards in your model (not addressed with stratification, inclusion of time-varying covariates or other technique to allow for nonproportional hazards), I suggest you consider presenting results in a logistic regression framework either in a sensitivity analysis or as the primary statistical analysis method. By refraining the reader from observing a survival plot (both an unadjusted Kaplan-Meier curve and a plot using the `stcurve` command (stratified by SFT) after fitting the Cox model in Stata), I find the correct interpretation of results and applicability really frail.

Response: We have added the multivariable logistic regression model as a sensitivity analysis (Supplementary Table 5 and 6) as suggested. The result was added in the Results section as follows; “We observed that constant proportional hazard assumption was not strongly hold in one variable (the modified SOFA score); therefore, additional sensitivity analyses were performed by using logistic multivariable models. The similar effect of SFT was observed (Supplementary Table 5 and 6).” Figures for unadjusted and adjusted Kaplan-Meier curves have been added as suggested.

6) Acknowledge in the limitations session that this is a retrospective cohort study still subject to confounding and that these results do not necessarily imply causation.

Response: We have added the limitation as follow in the discussion section; “Third, due to the use of observational data, the observed effects of the SFT on 28-day mortality in our study should be interpreted conservatively as an association rather than a causation.”

While I have made these criticisms, I reinforce the dataset is strong and should be published, but the statistical analysis methods and interpretation seem to need some improvement. Thank you for the opportunity to review your manuscript.

Response: We are grateful for your critiques and support, which helps us to improve this works.

All contributing authors have reviewed and concurred with the revised manuscript.

VERSION 3 – REVIEW

REVIEWER	Bruno Besen Medical ICU, Hospital das Clínicas, University of Sao Paulo Medical School, São Paulo (SP), Brazil
REVIEW RETURNED	03-Jan-2021

GENERAL COMMENTS	Dear authors, Congratulations for your interesting results. Kind regards, Bruno
--